# Learning Minimal Representations with Model Invariance

## Abstract

Sparsity has been identified as an important characteristic in learning neural networks that generalize well, forming the key idea in constructing minimal representations. Minimal representations are ones that only encode information required to predict well on a task and nothing more. In this paper we present a powerful approach to learning minimal representations. Our method, called MODINV or model invariance, argues for learning using multiple predictors and a single representation, creating a bottleneck architecture. Predictors' learning landscapes are diversified by training independently and with different learning rates. The common representation acts as a implicit invariance objective to avoid the different spurious correlations captured by individual predictors. This in turn leads to better generalization performance. We test MODINV on both reinforcement learning and vision settings, showcasing strong performance boosts in both. It is extremely simple to implement, does not lead to increased wall clock time while training, and can be applied across different problem settings.

## 1 Introduction

Learning efficient representations that generalize well is a long standing problem of machine learning, and particularly of deep learning (Doersch & Zisserman, 2017; Noroozi & Favaro, 2016; Oord et al., 2018). Algorithms that exploit structure in the real world through effective inductive biases are key to solving this problem. Several inductive biases have been successfully used in the past, from early work using translational invariance for developing CNNs (LeCun et al., 2004; 2010), to recent data augmentations for developing multiple self-supervised algorithms (Chen et al., 2020; Grill et al., 2020; Chen & He, 2021). A key inductive bias in multiple works is sparsity of concepts (Hoefler et al., 2021). Essentially, it refers to how objects in the real world tend to interact and affect the dynamics of other objects *only* in a small neighbourhood. Consider for example picking up a pen. This simple task only changes the dynamics of a very small part of the world, namely the pen and the hand of the person grasping it. Exploiting such structure allows learning representations that can generalize better in the real world.

In this paper, we develop a method that exploits sparsity by learning minimal representations. The well known saying 'neurons that fire together wire together' points to the phenomenon that that neurons that have similar output for a given input also have strong weights between them (Hebb, 2005). Ideally, given the same set of neurons to model certain concepts, we would want neurons that relate to a particular concept to have strengthened weights while those relating to another concept to have weakened weights (Sun et al., 2016). Therefore, low correlation in the outputs of the representation would lead to less redundancy in modelling concepts and thus better generalization. This is precisely the motivation behind minimality leading to sparse representations and improved generalization. Minimal representations (Tishby et al., 2000; Shamir et al., 2010) refer to ones that have sufficient but minimal information w.r.t. the task at hand. Therefore, it must be only possible to solve the task in hand and no other task, thus leading to better generalization for the given task. Another viewpoint is that minimal representations also lead to a reduction in spurious correlations. Since there is minimal noise present in the representation to cause spurious correlations, this again results in better generalization.

Our approach, which we call model invariance (MODINV), learns minimal representations by incorporating multiple predictors over a single, common representation for a given task (see Figure 1). Each predictor head is then trained independently (a given sample only trains one head) and with a learning rate different to other heads. Theoretically, this can be looked at as a manifestation of the information bottleneck principle, which balances between sufficiency and minimality. Intuitively, this can be looked at as providing robustness to spurious correlations. Since each head learns a different prediction (based on its own optimization and learning on different parts of data set), each head is susceptible to different spurious correlations. However, since there is only one common representation to support all the predictor heads, this representation must be robust to any spurious correlations as training progresses.

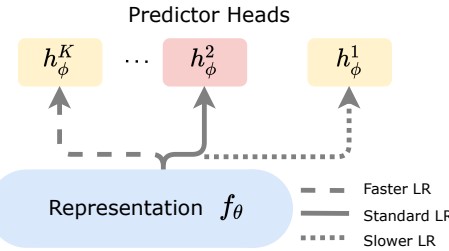

Predictor Heads

Figure 1: **Model Invariance (MODINV)** A common representation $f_\theta$ supports multiple predictor heads, each with a different learning rate. Each head is optimized based on the task objective (e.g., SSL loss) and captures different spurious correlations. As diversification in learning of predictor heads increases, the common representation must be optimal to reduce the loss for all predictors. 'Red' denotes the head being trained for the current sample.

We evaluate our method on two separate settings; **1) Reinforcement Learning:** The DeepMind Control Suite(DMC) (Tassa et al., 2018) benchmark with distractors, which involves natural video playing in the background while the agent is controlled in the foreground and **2) Vision:** The CIFAR-10 (Krizhevsky et al., 2009) and STL-10 (Coates et al., 2011) datasets, where no labels are used for learning the representations, followed by the standard linear probing evaluation protocol to test generalization.

**Contributions:** We introduce MODINV, a simple method motivated by incorporating sparsity in representations, while showing connections to the information bottleneck principle. MODINV is conceptually simple, easy to implement, and improves representation learning performance in both reinforcement learning (from pixels) and vision settings. In reinforcement learning, MODINV achieves stronger than state-of-the-art performance on the DMC Suite with distractors benchmark, both with augmentations and without. In vision, MODINV leads to better performance than the baseline method of SIMSIAM when using a linear predictor on both CIFAR-10 and STL-10 datasets.

## 2 PRELIMINARIES

### 2.1 INFORMATION BOTTLENECK PRINCIPLE

The well known information bottleneck (IB) principle (Tishby et al., 2000; Shamir et al., 2010) formally describes the ideas of sufficiency and minimality for learning representations. Specifically, it states that representations should contain sufficient information to do well on the downstream task but also contain minimal information about the inputs, leading to better, more robust generalization. This is in contrast to minimizing the empirical risk only, which is a sound technique in the infinite sample case. However, in the finite sample case, minimizing only empirical risk can lead to poor generalization. A similar observation has been noted in works that build over the invariant risk minimization principle (IRM) (Arjovsky et al., 2019).

### 2.2 SELF-SUPERVISED LEARNING IN RL

RL considers the agent's interaction with the environment as a discrete time $\gamma$-discounted Markov Decision Process (MDP) (Puterman, 2014) $\mathcal{M} = (\mathcal{X}, \mathcal{A}, P, R, \gamma, \mu_0)$, where $\mathcal{X}$ denotes a finite state space and $\mathcal{A}$ is the action space; $P \equiv P(x'|x, a)$ is the transition kernel; $R \equiv r(x, a)$ is the reward function; $\gamma \in [0, 1)$ is the discount factor; and $\mu_0$ is the initial state distribution. The objective is to find a policy $\pi : \mathcal{X} \to \Delta_{\mathcal{A}}$, where $\Delta_{\mathcal{A}}$ is the set of probability distributions on $\mathcal{A}$ such that the value function of a policy $\pi$ at a state $x \in \mathcal{X}$, $V^\pi(x) \equiv \mathbb{E}[\sum_{t \geq 0} \gamma^t r(x_t, a_t)|x_0 = x, \pi]$ is maximized. Recently a lot of progress has been made in making RL over pixels as sample efficient as when learning over true states. Most approaches can be divided into two classes. The first involves using auxiliary losses over the representation so as to inject as much information about the downstream task as possible. These auxiliary objectives include reconstructing observations (Hafner et al., 2020; Ha &

Schmidhuber, 2018), or predicting next observation, reward (Schaul et al., 2015) or even encoded states (Gelada et al., 2019). The second class borrows ideas from self-supervised learning methods and augments the observations (Yarats et al., 2021; Laskin et al., 2020b) with techniques like random crop, cutout, color jitter etc. Since randomly augmented samples from the same observation have the same $Q$ values, an implicit invariance to augmentations is enforced in the representation pipeline. In the case of hard exploration tasks, auxiliary objectives such as estimating value functions of random cumulants has shown to be useful as well.

A common baseline architecture used in a lot of methods is an actor-critic setup where a common convolutional encoder is used to compute the encoded latent state from the raw pixel observations. For continuous control tasks, soft actor-critic (SAC, Haarnoja et al. (2018)) is used as the actor-critic algorithm and the gradients from the actor are stopped from updating the encoded state. Only the critic trains the representation network $f_\theta$. This is since allowing gradients from both actor and critic leads to noisy estimates and the eventual divergence. Moreover, some methods also add a reward prediction loss from the encoded state, a transition prediction loss, or a pixel reconstruction loss.

### 2.3 SELF-SUPERVISED LEARNING IN VISION

An important property of self-supervised methods in vision is avoiding collapse in representation, or trivial solutions where the representation simply outputs a constant. These involve the use of two similar or identical networks, an online and a target network. Particular approaches largely fall in two categories, that of contrastive (Chen et al., 2020) and non-contrastive methods (Grill et al., 2020; Chen & He, 2021), depending on how they avoid representation collapse. Contrastive methods create positive and negative pairs from random augmentations of data (e.g., cutout, crop, flip), and enforce an objective such that positive samples are brought closer and negative samples are pulled away in representation space. On the other hand, non-contrastive methods do away with creating positive and negative pairs and instead use the likes of assymetric architectures, stop gradient, and momentum encoders for the target network to learn non-trivial representations.

One method we use to build over later in the paper is SIMSIAM (Chen & He, 2021), which is a non-contrastive method and only incorporates the most essential components required for preventing collapse. In particular, it uses two networks, an online and a target network, where both use a backbone network (usually a ResNet 18 or ResNet 50 (He et al., 2016)) and a projection MLP (1 hidden layer). Moreover, only the online network has a prediction MLP attached to it, which makes the overall architecture assymetric. The target network has a stop gradient applied to it and the overall objective is a cosine similarity loss between the outputs of the online and target networks. It has been noted that stop-gradient is enough to avoid collapse and that slowly updating the target network weights to match those of the online network, i.e. exponential moving average (EMA), is not a necessary requirement. Adding the EMA to the SIMSIAM architecture results in the BYOL (Grill et al., 2020) architecture.

## 3 METHOD

Consider a task $Y$, which can correspond to a variety of objectives. For instance, it could be an auxiliary task of predicting the next state when interacting with an RL environment, or simply a classification task on a vision benchmark. We are interested in training a model that can do well on task $Y$, where the model consists of two components, a backbone representation $f_\theta$ and a predictor/classifier $h_\phi$ which is attached over the representation. In a lot of tasks that involve pixel input, spurious correlations can arise due to various reasons such as lack of non-iid data, irrelevant information, confounders etc. that can result in poor generalization at test time. An attractive property for better generalization has been the idea of sparse representations, in that the mutual information of any two dimensions of the representation must be low. However, in order to avoid learning trivial solutions, the representation as a whole should encode enough information about the downstream task as well. A combination of both of these objectives then leads to a model with better generalization capability. In this paper, we balance the above two objectives using the idea of model invariance (MODINV). MODINV deploys multiple predictors over a single representation, while training each predictor independently, each capturing different spurious correlations. The common representation acts as a implicit invariance loss which ensures that only the optimal representation remains at convergence. Intuitively, each predictor can be looked at as an augmented version of the optimal

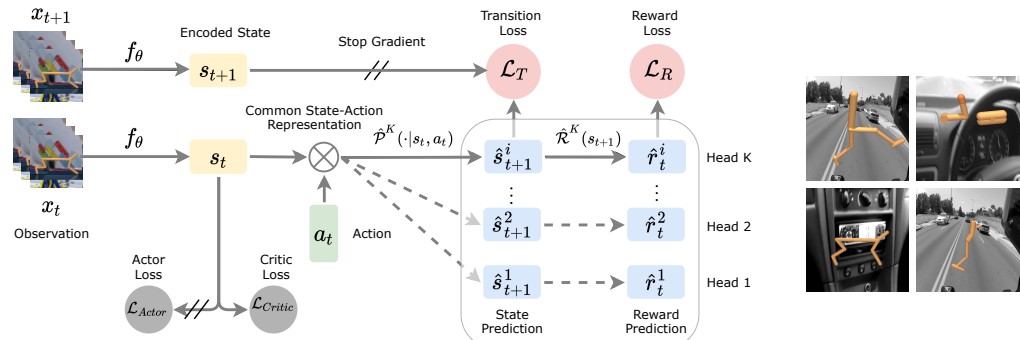

Figure 2: **MODINV for Reinforcement Learning**. **Left**: The key idea is to use multiple models of the next encoded state, each grounded with its own reward decoder. At any training iteration, only one of the heads is trained for the sampled batch of observations. Each head is also independently initialized and uses a different learning rate w.r.t. the other heads. **Right**: Natural Distractor in the background for different tasks from DMC Suite.

model, where the augmentation is over the model space and refers to particular spurious correlations arising in the model that differ it from the optimal model. Note that each predictor head is the same in architecture, and we only diversify the learning procedure of each head (through different initialization, independent training, different learning rates). This is so that we eventually converge to an optimal representation for a particular predictor/classifier and not for all predictor/classifier families, which is a much more hard to optimize in practise. For all experiments, we use a random routing to decide which data sample is used to train which predictor head. Furthermore, the learning rates of the different predictor heads are chosen intuitively, i.e. if the base rate is 3e-3 then we choose one slightly higher rate (5e-3) and one slightly smaller rate (1e-3) for K=3 predictors.

## 4 EXPERIMENTS

### 4.1 REINFORCEMENT LEARNING

For RL over pixels, we build over a standard actor-critic setup and use MODINV as part of an auxiliary task that predicts the next latent state (see Figure 2). This auxiliary task is used to train the representation which supports the actor-critic heads. Specifically, the input is the observation $x_t$ and the representation $f_\theta$ encodes it into the latent state $s_t$. We then concatenate the latent state and the action and pass it through a linear MLP to generate a common state-action representation (predicting the next state $s_{t+1}$ requires information about both the state and action). Since the true latent state is not known, predicting only the next state $s_{t+1}$ can result in a representation collapse[*]. Therefore, we ground the state prediction loss by also predicting the reward, which does have a valid grounding and thus avoids collapse. Finally, we deploy MODINV by using multiple predictor heads for the state prediction task, with each head being coupled with a corresponding reward prediction head.

### 4.1.1 IMPLEMENTATION DETAILS

We implement SAC (Haarnoja et al., 2018) as the base agent with the actor and critic sharing a common representation. Our actor-critic setup is similar to the one used in SAC-AE (Yarats et al., 2019), except the reward and state prediction task. Our transition or state prediction model is deterministic in nature, involving a 3 layer encoder MLP, followed by a 3 layer decoder MLP. The reward prediction network is a 2 layer MLP. We do not tune for MLP sizes and widths at all in reporting our results. Both the reward and transition losses train the representation alongside the critic. We use three MODINV heads, each with a slightly different learning rate of $3e − 3, 3e − 4$, and $3e − 5$ respectively.

---

[*]Recently it was shown that collapse can be prevented when using a cosine loss and an architecture similar to BYOL instead of a squared error loss.

### 4.1.2 RESULTS

We test our method on the DMC Suite with distractors. Particularly, we test on the six popular domains of Walker Walk, Reacher Easy, Hopper Hop, Finger Spin, Cheetah Run and Cartpole Swingup. We compare with other methods that learn over pixels including DREAMER, CURL, DBC, and RAD. Dreamer (Hafner et al., 2020) is a model-based method that performs pixel reconstruction and reward prediction to learn a model in the latent state space and then performs model-based updates to a base SAC agent. CURL (Laskin et al., 2020a) uses a contrastive loss similar to that in CPC (Oord et al., 2018) to train the representation network, using data augmentations to generate positive and negative samples. RAD (Laskin et al., 2020b) is another method that simply augments the samples in the replay buffer without adding any other loss function to the base SAC setup. Finally, note that both RAD and DrQ (Kostrikov et al., 2020) are similar in performance as they both use data augmentations in the replay buffer samples. Since data augmentations explicitly remove the actual distractions in the background, a direct comparison with data augmentations is not fair. Therefore, we also compare our method when augmentations are added alongside MODINV (denoted by MODINV+AUG). We evaluate all methods at both 100K and 500K environment steps, which is the standard number of steps in this benchmark.

Our results (Table 1) show that MODINV clearly outperforms all 4 methods (DREAMER, CURL, DBC) at the 500K step mark. Note that since all methods use the same number of gradient steps, MODINV is at a slight disadvantage as the total number of gradient updates are divided amongst the $K$ predictor heads.

## 4.2 VISION

For vision, we adopt the base framework of SIM-SIAM, a non-contrastive self-supervised method for learning representations. SIMSIAM has three model components, the backbone network or the representaion, a projection MLP and a prediction MLP. We deploy MODINV in the projection MLP, with a single backbone representation and a single prediction MLP (see Figure 3). At each training iteration, the same MODINV projection head is sampled for both the online and target networks and trained using the original SIM-SIAM loss function. Unlike the reinforcement learning case, where any of the modules after the representation (state and reward prediction

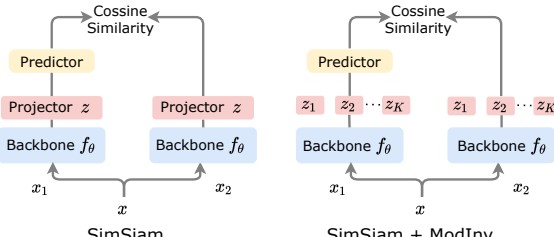

Figure 3: **MODINV for Vision**. The key idea is to use multiple projectors for both the online and target networks. At any training iteration, only one of the heads is trained for the sampled batch of data. Each head is also independently initialized and uses a different learning rate w.r.t. the other heads.

heads) are not used actively in the algorithm, in the vision case the projector and predictor both are used to align the backbone representation (Tian et al., 2021). Therefore, ensuring that each projection head in MODINV is trained sufficiently (as in the base case) is important.

### 4.2.1 IMPLEMENTATION DETAILS

We test our approach on the CIFAR-10 and STL-10 datasets. As data augmentations, we use random crop, color jitter, horizontal flip, random grayscale for both datasets and also use blur for STL-10. Furthermore, we deploy three MODINV heads for the projection MLP, with the learning rates of $0.03, 0.3, 0.003$ for each head respectively. The learning rate of all other components in the model is set to $0.03$, with a weight decay of $0.0004$, momentum $0.9$. We run two sets of experiments, one with a linear prediction MLP and one with the standard two layer prediction MLP. The rest of the architecture is the same as that of SIMSIAM, with the output dimension of the projector and the predictor MLP set to 2048. We use a batch size of 512 for CIFAR-10 and 128 for STL-10.

**Linear *vs* Non-linear Predictor** . For the SSL setting we deploy MODINV in two different configurations. The first involves a linear predictor while the second involves a standard 2-layer predictor. The 2-layer predictor has been speculated to be using a lot of 'lucky' initialization that lead to alignment between the backbone representation and the prediction MLP (Tian et al., 2021). However, the linear predictor does not have this property and thus testing in this setting offers a much more

Table 1: **Comparisons on DMC Suite + Distractors**. Performance of different methods at the **100K** and **500K** mark. We report mean and std deviation for 5 seeds. MODINV is compared with DREAMER (Hafner et al., 2020), CURL (Laskin et al., 2020a), DBC (Zhang et al., 2020), and RAD (Laskin et al., 2020b). MODINV without augmentations uses a `stopgrad` for the critic gradients while MODINV+AUG allows gradients, similar to RAD.

| 100K STEP SCORES | DREAMER | CURL | DBC | MODINV | RAD | MODINV+AUG |
|---|---|---|---|---|---|---|
| WALKER, WALK | $345 \pm 106$ | $60 \pm 36$ | $198 \pm 85$ | $246 \pm 76$ | $465 \pm 32$ | $371 \pm 76$ |
| HOPPER, HOP | $0.6 \pm 1.2$ | $0 \pm 0$ | $0 \pm 0$ | $0 \pm 0$ | $0 \pm 0$ | $0 \pm 0$ |
| CHEETAH, RUN | $101 \pm 53$ | $46 \pm 22$ | $48 \pm 9$ | $145 \pm 54$ | $228 \pm 106$ | $148 \pm 66$ |
| REACHER, EASY | $56 \pm 85$ | $186 \pm 86$ | $202 \pm 51$ | $216 \pm 141$ | $482 \pm 261$ | $635 \pm 333$ |
| FINGER, SPIN | $188 \pm 86$ | $51 \pm 24$ | $170 \pm 137$ | $240 \pm 122$ | $782 \pm 31$ | $591 \pm 138$ |
| CARTPOLE, SWINGUP | $133 \pm 47$ | $254 \pm 18$ | $311 \pm 74$ | $567 \pm 98$ | $845 \pm 14$ | $668 \pm 64$ |
| **500K STEP SCORES** | | | | | | |
| WALKER, WALK | $691 \pm 174$ | $51 \pm 16$ | $572 \pm 120$ | $742 \pm 99$ | $930 \pm 19$ | $870 \pm 43$ |
| HOPPER, HOP | $8 \pm 14$ | $0.9 \pm 0.5$ | $1.2 \pm 1$ | $\mathbf{25} \pm \mathbf{50}$ | $112 \pm 96$ | $65 \pm 65$ |
| CHEETAH, RUN | $266 \pm 73$ | $50 \pm 19$ | $275 \pm 71$ | $\mathbf{462} \pm \mathbf{115}$ | $425 \pm 33$ | $361 \pm 59$ |
| REACHER, EASY | $223 \pm 384$ | $199 \pm 137$ | $118 \pm 41$ | $\mathbf{552} \pm \mathbf{197}$ | $935 \pm 4$ | $976 \pm 3$ |
| FINGER, SPIN | $387 \pm 123$ | $34 \pm 12$ | $755 \pm 67$ | $780 \pm 189$ | $982 \pm 4$ | $974 \pm 13$ |
| CARTPOLE, SWINGUP | $119 \pm 37$ | $199 \pm 65$ | $626 \pm 80$ | $\mathbf{814} \pm \mathbf{11}$ | $837 \pm 23$ | $837 \pm 6$ |

Table 2: **Comparisons on STL-10 and CIFAR-10**. All are based on **ResNet-18** pre-training. Evaluation is on a single crop.. MODINV is added to the base architecture of SIMSIAM (Chen & He, 2021) with 2-layer projector (2048-d) and 2 layer predictor with a 512-d hidden layer. We use weight decay of $5e - 4$, learning rate 0.03 with a cosine decay.

| | linear predictor | STL-10 | | | | CIFAR-10 | | | |
|---|---|---|---|---|---|---|---|---|---|
| | | 40 ep | 60 ep | 80 ep | 100 ep | 50 ep | 100 ep | 200 ep | 300 ep |
| SIMSIAM | ✓ | 77.1 | 80.7 | 83.7 | 85.1 | 40.5 | 33.0 | 41.4 | 41.3 |
| SIMSIAM + MODINV | ✓ | 77.0 | 78.4 | 81.2 | 83.5 | **51.7** | **70.7** | **78.9** | **82.5** |
| SIMSIAM | | 77.4 | 80.9 | 84.1 | 86.4 | 67.4 | 76.6 | 81.7 | - |
| SIMSIAM + MODINV | | 78.9 | 78.9 | 83.4 | 85.4 | 64.7 | 75.4 | 79.9 | - |

robust evaluation option. Unless specified otherwise, our ablations for vision in Section 5 use the linear predictor setting for MODINV.

### 4.2.2 RESULTS

We see that with a linear predictor, the base SIMSIAM version fails for CIFAR-10 and only achieves 41.3%[†] accuracy while the MODINV version achieves an accuracy of 82.5% at 330 epochs, similar to that of the 2-layer predictor case. A similar gap in performance is observed for earlier training epochs as well (Table 2). A careful reader might attribute this gap to the use of different learning rates in MODINV, since a relatively slower learning rate for the non-predictor layers can correspond to improved performance, especially when EMA is not used in the target network (Tian et al., 2021). To test this, we train the base SIMSIAM model individually with the different projector learning rates used in the MODINV version. We see that all three learning rates fail to the same performance of around 40% accuracy. This clearly shows that MODINV contributes much more to better generalization than using slower or faster learning rates.

---

[†]Note that this is not complete collapse since the accuracy still increases very gradually after 200 training epochs.

## 5 ABLATIONS

### 5.1 ABLATION ON NUMBER OF HEADS

In our experiments, we observe that only $K = 3$ heads are sufficient to provide good performance boosts compared to when not using MODINV (i.e. $K = 1$) across the RL and SSL settings. Furthermore, with everything else the same, adding an extra head after $K = 3$ leads to diminishing performance gains. This can be most clearly seen for SIMSIAM + MODINV, where we see increasing gains with increasing $K$ (see Table 3). Note that although relative gains saturate after $K = 3$, it is possible that further diversification in predictor training can lead to even better results when $K > 3$.

Table 3: **Ablation on CIFAR-10**. Performance of **100 epoch** pre-training for different number of heads.

|  | 1 head (SIMSIAM) | 2 heads | 3 heads | 5 heads |
|---|---|---|---|---|
| MODINV | 40.5 | 57.0 | 70.7 | 70.2 |

### 5.2 CORRELATION IN DIMENSIONS

As mentioned, our initial motivation stems from sparsity in concepts. This refers to low mutual information between any two dimensions in the representation $f_\theta$. A useful metric to check for such a characteristic is therefore the mean correlation between any two dimensions. We plot this metric during training with and without MODINV. Ideally, we would hope to get lower mean correlation when using MODINV as opposed to when not. Our experiments show that this is indeed the case, as the mean correlation decreases much more steadily when we use MODINV. Moreover, the decrease in this metric has a strong correlation with the performance of the agent.

$$\mathcal{L} = \sum_b \left( \hat{\mathcal{P}}(z_b) - \mathcal{P}(z_b) \right)^2 + \lambda \sum_i \sum_{j \neq i} \mathcal{C}_{ij}^2 \qquad \text{where } \mathcal{C}_{ij} = \frac{\sum_b z_{b,i} \, z_{b,j}}{\sqrt{\sum_b (z_{b,i})^2} \sqrt{\sum_b (z_{b,j})^2}} \qquad (1)$$

Based on this observation, we also run an experiment where we take the baseline architecture (i.e. without MODINV) and add the mean correlation between dimensions of $f_\theta$ as an auxiliary loss. Equation 1 describes the exact loss, where the summation is over the batch $b$. The first term is the state prediction error from Section 4.1 while the second term is the exact loss from Zbontar et al. (2021). Our results show that optimizing directly for this loss leads to much better performance in the Reacher Easy task, while much worse performance in the Walker Walk (Table 4). We choose to test on these two domains since they are hard to solve for most methods that we compare to. This is an interesting observation since similar losses that have been used successfully in vision recently (Bardes et al., 2021) might offer strong performance improvements in RL as well.

Table 4: **Ablation for Mean Corr. Loss**. Performance at **500K** for MODINV and the mean correlation loss version from Eq. 1 for different $\lambda$ values.

|  | MODINV | Mean Corr. Loss (Eq. 1) | | | |
|---|---|---|---|---|---|
|  |  | $\lambda = 5e-3$ | $\lambda = 5e-4$ | $\lambda = 5e-5$ | $\lambda = 5e-6$ |
| WALKER, WALK | $742 \pm 99$ | collapse | $392 \pm 181$ | $479 \pm 227$ | $688 \pm 78$ |
| REACHER, EASY | $552 \pm 197$ | collapse | $713 \pm 346$ | $739 \pm 201$ | $678 \pm 293$ |

### 5.3 IMPORTANCE OF DIFFERENT LEARNING RATES

Table 5: **Ablation for different learning rates for MODINV**. SIMSIAM + MODINV with linear predictor with different sets of learning rates for the 3 projector heads.

|  | {3e-2, 3e-1, 3e-3} | Same lr | {3e-2, 3, 3e-4} |
|---|---|---|---|
| MODINV | 70.7 | 50.7 | 52.0 |

When using multiple heads in MODINV, it is important to diversify the learning of heads so they capture different spurious correlations. Using different learning rates is a vital component for achieving this, besides different model initialization and independent training. We test this by evaluating performance with multiple sets of learning rates for $K = 3$ predictor heads. We see that performance improves when the learning rates are diverse as compared to when all heads use the same learning rate (Table 5). Moving on to diversifying learning of predictor heads through different optimizers is left for future work (see Section 7).

## 5.4 ABLATION ON ONLY PREDICTING REWARD

The RL application of MODINV involves both state prediction and reward prediction. We now test how the performance might vary if we only have the reward prediction task. It is worth noting that MODINV can be applied to the actor-critic heads as well, however optimization in that case becomes difficult as the critic learning loss does not have a stable ground truth. In Table 6 we see that performance deteriorate in general for the three tasks, especially for Cheetah Run. This may suggest that state prediction provides a slightly more stable ground truth to optimize the representation.

## 5.5 ABLATION WITH DATA AUGMENTATIONS

Our results show that MODINV acts complimentarily to augmentations. In our experiments in the RL setting, we observe that adding data augmentations over MODINV leads to better performance than when only using MODINV. 'Crop' is the only augmentation used here, which is standard for RL environments. Essentially, cropping removes a lot of background information which is irrelevant to the task (top-down approach), thus aiding in better or more robust

Table 6: **Ablation on MODINV for RL**. Performance at **500K** for only state prediction vs both state and reward prediction.

|  | Only Reward | State and Reward |
|---|---|---|
| WALKER WALK | $704 \pm 85$ | $742 \pm 99$ |
| REACHER EASY | $543 \pm 271$ | $552 \pm 197$ |
| CHEETAH RUN | $235 \pm 83$ | $462 \pm 115$ |

estimates for next state and critic losses. However, not all augmentations have a similar effect (e.g. flip), and thus end up with similar performance as the baseline. On the other hand, MODINV avoids spurious correlations by enforcing minimality, resulting in a bottom-up approach to robust reward and critic loss optimization.

# 6 RELATED WORK AND DISCUSSION

## 6.1 DECODABLE INFORMATION BOTTLENECK

The IB principle does not take into account the classifier/predictor family which is attached over the representation and thus can be considered too restrictive. Moreover, the minimality term is hard to approximate for practical losses. To remedy this, the decodable information bottleneck (Dubois et al., 2020) recently introduced notions of sufficiency and minimality for a given predictor/classifier family. They also provided a practical method which has a similar structure to the MODINV approach. In particular, multiple predictor heads are attached to a common representation, where one head (sufficiency head) minimizes the standard empirical risk while all other heads (minimality heads) are learnt such that they cannot predict arbitrary relabellings of the same data. This is done by providing different random relabellings and then reversing the gradients for all the minimality heads. This ensures that the common representation contains information required to predict only the correct labels and no other. The idea in MODINV is similar but we do not deploy reverse gradients or relabel data, which in practise can lead to instability issues. Therefore, MODINV can be seen as a specific instantiation of the DBC framework, following similar theoretical guarantees while being much easier to implement and resulting in better performance.

## 6.2 DECORRELATION IN NON-CONTRASTIVE SSL

Recently, BARLOW TWINS (Zbontar et al., 2021) was introduced as a new non-contrastive self-supervised learning method which does not use an asymmetric architecture while still avoids collapse in representation. This was achieved by enforcing the off-diagonal elements of the cross-correlation

matrix of the outputs of the representation to be zero. Essentially, such a loss ensures that each output dimension of the representation is decorrelated with each other while minimizing the standard loss of enforcing invariance to augmentations. Interestingly, the BARLOW TWINS loss can be traced back as an approximation to the information bottleneck principle again, hence connecting decorrelation in representation dimensions as an indicator for better generalization (as noted similarly in the introduction). We showed that MODINV also leads to more decorrelation in the representation dimensions, thus supporting this observation. However, we also showed that directly optimizing for this loss might not be a good idea always, with such a variant leading to worse empirical performance than MODINV in the RL setting. Nevertheless, it is important to note that the MODINV and BARLOW TWINS objectives are primarily complimentary in nature. This is since MODINV always requires a sufficiency objective to start with, which could be provided by any SSL loss which avoids collapse, be it SIMSIAM with the use of a predictor and stop gradient or be it BARLOW TWINS with the use of the decorrelation loss. In similar flavor to BARLOW TWINS, VICREG (Bardes et al., 2021) also uses the decorrelation loss from BARLOW TWINS and adds a variance regularization loss to prevent collapse. Finally, Hua et al. (2021) introduced a technique called decorrelated batch normalization (DBN) and also showed that collapse can be avoided with no predictor.

### 6.3 AUXILIARY VALUE PREDICTION IN RL

To improve representation learning in RL, a prominent idea has been that of predicting multiple value functions from a common representation. Sutton et al. (2011) use random cumulants to define the different value functions. The value improvement path (Dabney et al., 2020) idea uses a mixture of value functions of past policies. Although these works have a similar looking structure to MODINV, there are a couple of critical differences. First, the multiple heads in MODINV are predictors of the same object while the above papers use different objects (different value functions). Second, the training of each head is not independent. These two differences dissociate the above works from the IB idea, thus making them complimentary to MODINV.

## 7 FUTURE WORK

MODINV shows promising improvements in both the RL and SSL settings, thus hinting at the benefits offered by the IB principle in practice. Nevertheless, MODINV is still one instantiation of this idea. There remain quite a few directions to explore in future. In particular, analysing how further diversification in the $K$ predictors can be achieved is a promising idea. For example, we can introduce different optimizers (Adam, SGD) for each predictor, while also varying parameters such as momentum, weight decay etc. Another possibility could be to update *a subset* of predictors for a given sample, instead of only updating *one* predictor at each training iteration. Finally, since data augmentations change the input data, each predictor could be trained on a different augmentation scheme, thus ensuring further diversification in optimizing different predictors.

On a separate axis, analysing how similar MODINV for vision is with other SSL methods such as VICREG is interesting. For the RL case, MODINV can also potentially be applied to the Q functions directly (each predictor is a separate Q head over a common representation) instead of the reward and transition modules. This could turn out to be hard to optimize for certain environments though, since Q functions can be less developed for sparse reward settings.

## 8 CONCLUSION

We introduce a general representation learning method motivated from the ideas of sparsity and minimal-sufficient representations. Our method, MODINV, uses multiple predictors over a common representation, diversifying the training of each predictor such that the common representation acts as an implicit invariance mechanism to spurious correlations. We show instances of this idea in both reinforcement learning and vision settings. Overall, MODINV leads to more robust generalization across different evaluations, leading us to believe that with further analysis, we can get further gains in representation generalization. MODINV can be viewed as a specific instantiation of the information bottleneck principle, one that is simple in implementation and in concept.

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

# Appendix

## 1 HYPERPARAMETERS

Table 7: **Hyperparameters** for MODINV for RL.

| Hyperparameter | Values |
|---|---|
| Observation shape | (84, 84, 3) |
| Latent dimension | 50 |
| Replay buffer size | 100000 |
| Initial steps | 1000 |
| Stacked frames | 3 |
| Action repeat | 2 finger, spin; walker, walk |
| | 8 cartpole, swingup |
| | 4 otherwise |
| SAC: Hidden units (MLP) | 1024 |
| Transition Network: Hidden units (MLP) | 128 |
| Transition Network: Num Layers (MLP) | 6 |
| Reward Network: Hidden units (MLP) | 512 |
| Reward Network: Num Layers (MLP) | 3 |
| Evaluation episodes | 10 |
| Optimizer | Adam |
| $(\beta_1, \beta_2) \rightarrow (f_\theta, \pi_\psi, Q_\phi)$ | (.9, .999) |
| $(\beta_1, \beta_2) \rightarrow (\alpha)$ | (.5, .999) |
| Learning rate $(f_\theta, \pi_\psi, Q_\phi)$ | 2e-4 cheetah, run |
| | 1e-3 otherwise |
| Learning rate $(\alpha)$ | 1e-4 |
| Batch Size | 128 |
| Q function EMA $\tau$ | 0.005 |
| Critic target update freq | 2 |
| Convolutional layers | 4 |
| Number of filters | 32 |
| Non-linearity | ReLU |
| Encoder EMA $\tau$ | 0.005 |
| Discount $\gamma$ | .99 |

