# OpenReview forum: "Learning Minimal Representations with Model Invariance"
_ICLR.cc/2022/Conference — ICLR 2022 Submitted_

### Official Review · Reviewer_LtXH · 2021-10-26

**Correctness:** 1
**Technical Novelty And Significance:** 1
**Empirical Novelty And Significance:** 1
**Recommendation:** 3
**Confidence:** 4

**Main Review:**

**Strengths:**
- Simple method -- just train 3 heads with different learning rates.
- No increase in wall-clock time
- Clear writing

**Weaknesses:**
- **No evidence for the main claims**
    - *”Common representation acts as an implicit invariance objective to avoid the different spurious correlations captured by individual predictors”:* There is no evidence of this, i.e. that the proposed method better deals with (or avoids) certain spurious correlations.
    - *Each head “captures different spurious correlations”:* Again, there is no evidence for this claim.
    - *”[...] diversify the learning of heads so they capture different spurious correlations”* Again, no evidence, e.g. an analysis of what correlations are captured by each head.
    - *”[...] This in turn leads to better generalization performance”:* There is no evidence that the proposed method achieves better generalization performance by avoiding spurious correlations. If in-distribution test-set performance is what is meant by generalization performance, then I think this needs to be made clear. Nonetheless, to justify the original statement, it would still need to be shown that this improvement is due to the avoidance of spurious correlations.
- **Questionable claims**:
    - *Training independently*: the predictors are given the same samples in the same order, and even share the same (backbone) representation. “Trained independently” is a bit of a stretch…
    - *”Strong performance boosts in both [RL and vision settings]”:* It seems that the proposed method ModInv only improves performance when additional training stability is needed (perhaps unsurprising since it's an ensemble method).
      - *RL:* With data augmentation to already stabilize training, ModInv only improves performance on 1 of 6 tasks over RAD, decreasing performance on 4 of 6 tasks (500k steps).
      - *Vision*: ModInv only helps when using a linear predictor on CIFAR-10. When using a nonlinear predictor on CIFAR-10 (which is the standard and improves performance with or without ModInv) or any predictor on STL-10, ModInv actually hurts performance.

**Other comments:**
- **Missing results**: Section 5.2 -- where are the results mentioned that show the mean correlation decreasing more steadily with ModInv?

**Summary Of The Paper:**

This paper proposes using an ensemble of predictors or heads on top of a shared representation to improve performance. The ensemble of predictors/heads are made "diverse" by using different learning rates. Some performance gains are shown in RL settings, while some stability gains are shown in a particular vision setting.

**Summary Of The Review:**

While the method is admirably simple and the writing clear, the main claims of the paper are either questionable or without any supporting evidence. Thus, in my opinion, this paper falls well short of acceptance for a top-tier conference like ICLR. If evidence can be provided in support of the main claims, I would be happy to revise my score.

---

> ### Author Response · Authors · 2021-11-23
> **Author Response**
>
> Thank you so much for your review and the constructive comments! Please find our detailed response below:
>
> **On the main claims**: We have added a toy experiment to show how/if ModInv removes spurious correlations. This is explained in the common response. We have also added ImageNet results which show improved generalization. You are correct in saying that by generalization we refer to in-distribution test-set performance, as is the case for the vision results. For the RL case, there are no explicit test-sets so we consider the final reward as a proxy for how well the agent is generalizing (with all other algorithmic details remaining same for the base model and the one with ModInv added), as is standard practice.
>
> **"Training independently"**: We actually do sample one predictor at each training iteration, and only train one of the predictors using that sample. So training is as independent as it can be without deploying more sophisticated techniques to segregate data samples.
>
> **"Strong performance boosts in both [RL and vision settings]"**:
>
> **RL**: We found a bug in the modinv + aug results for the RL experiments. As noted in the main comment, fixing these results in most scores remaining within 1 standard deviation of each other. Since the maximum score for DMC envs is 1000, there is very little ceiling for improvement of results with augmentations added for most tasks.
>
> **Vision**: For the self-supervised vision experiments, the asymmetry of the base SimSiam architecture requires that the asymmetric predictor “keeps up” with the projector. Since in modinv we are using multiple projectors, the predictor is trained for all samples but only one projector is trained for a particular sample. This is why we see that the performance slightly lags with the base case. We do not see this for a symmetric architecture like Barlow Twins (BT), as is observed by our ImageNet results. In the future, testing ModInv alongside a symmetric architecture such as SimCLR would make more sense, particularly because it does not use a sparsity inducing term like in Barlow twins which might diminish the gap between vanilla BT and BT + ModInv

---

> > ### Comment · Reviewer_LtXH · 2021-11-27
> > **Response to authors**
> >
> > Thank you for your response. I take your point about sampling a predictor on each iteration—I missed that detail and it does make them less dependent. However, I retain my score as:
> > - The main claims remain without evidence
> > - The results remain unconvincing
> > - Missing results not addressed

---

### Official Review · Reviewer_Q19W · 2021-10-31

**Correctness:** 3
**Technical Novelty And Significance:** 2
**Empirical Novelty And Significance:** 3
**Recommendation:** 6
**Confidence:** 4

**Main Review:**

## Strengths

- The paper is well-written and easy to understand. The intuitions and connections to relevant related works have been established meaningfully.
- The ablations are very well done, and reasonably support many of the claims.

## Weaknesses

I have two main concerns to be fully confident of the gains claimed.

- The proposed scheme lies very close, in principle, to a family of approaches called the [Hyperparameter Ensembles](https://arxiv.org/abs/2006.13570) (HE). HE induce diversity by training using different hyperparameters. The obvious difference is the fact that the authors here only train the predictor heads in an HE manner instead of the full network. Nevertheless, approach is very closely related.

  Here, the authors claim the benefits from an information bottleneck perspective, but the intuitions only remain justified empirically. In that sense, I would then believe that the gains are due to the same reason deep ensembles work, than for an information-theoretic reason of learning minimal representations.

  The fact that adding more than $K=3$ heads leads to diminishing returns may also hint to qualitative similarity to deep ensembles (where an ensemble beyond $\sim 5$ leads to diminishing returns).

- Another intuition that the authors highlight is that forcing the same representation induces an information bottleneck. This is turn should nudge the network away from learning spurious correlations. This claim is hand-wavy and is not supported either empirically or theoretically (perhaps even a toy example would suffice to qualitatively guarantee that it is doing what the authors claim it to be doing). My impression here is again that it is not that the minimal representations are learned that help avoid spurious correlations and hence better generalizations. It is instead the same reason that deep ensembles work, i.e. finding a set of good minima which induce a reasonable functional diversity. Deep ensembles are not very robust to spurious correlations either (though better than a single network), and it is unlikely that a very similar training routine proposed in this paper does something very different.

Nevertheless, I do want to acknowledge that the setting of using RL with pixels and the achieved gains is still a potentially novel intersection. But the authors approach the problem from an information bottleneck perspective, the robustness claims from which are not entirely supported.

## Minor

- Highlighting best results in Table 1 would make it easier for the reader.
- Hyperlinks do not work in the main submission.



**Summary Of The Paper:**

Inspired by the information bottleneck principle, the authors propose to learn minimal representations by independently training multiple predictive heads on top of a shared representation, such that the resultant predictor is more robust to spurious correlations. To test this proposal, the authors provide extensive set of experiments in RL from pixels and vision benchmark tasks. The ablation studies help make the case too.

**Summary Of The Review:**

The main theme of my concerns is that it is not that the training has learned minimal representations but just that it is the diversity of predictor heads that provides the gains. Without sufficient empirical or theoretical evidence, I think the main claim of the paper of being robust to spurious correlations due to the implicit information bottleneck is weak.

---

> ### Author Response · Authors · 2021-11-23
> **Author Response**
>
> Thank you so much for your review and the constructive comments! Please find our response below:
>
> Thank you for pointing to the Hyperparameter Ensembles idea. It is certainely very interesting. However, note that the common representation is usually much, much bigger than the K decoders/predictors. For ex. even for the CIFAR-10 experiments, the representation is a ResNet-18 model while the predictors are linear models or 1-layer MLPs. Therefore, in terms of capacity, ensembles would be very different than the architecture proposed here.
>
> Nevertheless, to your point about spurious correlations, we have tested our minimality claim in a toy supervised learning setup (shown in the common response). This shows that ModInv is removing spurious correlations more than the base model. Of course, when we use a finite number of predictors (3 in this case), we only get minimality approximately. Therefore, more engineering that explores how to make the best out of minimum number of predictors is an interesting future direction. We hope the toy experiment addresses your concerns on claiming minimality with ModInv.
>
> Minor:
>
> - We have highlighted the main results in Table 1 as well (those upto statistical significance).
> - The updated manuscript has the hyperlinks working now.

---

### Official Review · Reviewer_vnKu · 2021-11-02

**Correctness:** 1
**Technical Novelty And Significance:** 2
**Empirical Novelty And Significance:** 2
**Recommendation:** 3
**Confidence:** 4

**Main Review:**

Strengths:
* The idea in itself is interesting and can be applicable on top of most of self supervised algorithms.
* The authors evaluate their method on two very different sets of tasks (RL and Vision).
* The paper is well-written easy to read.

My main concern with the paper is its experimental soundness. I would strongly encourage the authors to resubmit a new version with more thorough experiments in order to back the (potentially interesting) claims that they make.

Weaknesses:
* From the elements currently provided in the paper, I don't see why the authors' algorithm would lead to minimal representation, at least in the information-theoretical sense as defined in the paper they cite: Shamir et al., 2010. I would expect either a citation or some experiments to support this claim.
  * For instance: "The common representation acts as a implicit invariance loss which ensures that only the optimal representation remains at convergence. " -> I need either a citation or evidence for that.
  * Similarly: "since there is only one common representation to support all the predictor heads, this representation must be robust to any spurious correlations as training progresses.".

In particular I don't see why each predictor would not learn to filter out unnecessary information in the representation. Using the paper's example of the pen: knowing the position of a distractor object is not necessary to solve the task, yet all the predictors could learn to filter out this useless information. In this case I don't see why the proposed algorithm would incentivize to remove this information from the representation.

* The authors claim "Our experiments show that [...] the mean correlation decreases much more steadily when we use MODINV."

Are these results in the paper? If not, adding them would benefit the paper a lot. The current Table 4. only shows that encouraging decorrelation of features enables better RL. It does not show that using ModInv provides better decorrelation of representation features.

* The author choose to evaluate in the specific case of a linear predictor. They claim :
  * "2-layer predictor has been speculated to be using a lot of 'lucky' initialization".
  * "testing in this setting offers a much more robust evaluation option".

These claims lack either citations or experiments.

Evaluating with a linear predictor is problematic for me as this case is not the standard regime. SimSiam is tuned for the 2-layer predictor regime and does not work well in the linear regime.I would expect the authors to either provide improvement in the 2-layer regime or convincing experiments or citations to support this choice.

* Can you clarify : "Unlike the reinforcement learning case, where any of the modules after the representation (state and reward prediction heads) are not used actively in the algorithm, in the vision case the projector and predictor both are used to align the backbone representation (Tian et al., 2021). "
For me it seems that in both cases the predictor and projector are used as tools to learn a good backbone representation. Even in the vision case the predictor and projector are thrown away when the network is actually used (for classification for instance).

Typo: "the phenomenon that that neurons that have"

**Summary Of The Paper:**

The authors propose MODINV, a new algorithm to learn minimal and compressed representations.
MODINV uses several predictors with different initializations and learning but all built on top of the same learnt representation.

After explaining the algorithm and its relation to minimal representation the authors evaluate it on vision and RL tasks.

On vision tasks the authors show improvement on CIFAR-10 by adapting their method on top of SimSiam in the specific case of a linear predictor. On RL they evaluate on control tasks from pixel inputs.

The authors also provide an experiment to show that adding a loss to discourage correlation within features helps to improve RL performance.

**Summary Of The Review:**

I think a lot of claims are not justified either by experiments or citations. Especially the main claim that the authors' algorithm learns a minimal representation.

I also found the vision experiments not convincing: on CIFAR-10, they used SimSiam in the linear prediction case instead of the usual 2-layer predictor in which SimSiam performs best.

From the elements provided in the paper, I could not tell whether the proposed algorithm was indeed helping to learn minimal representations, or that such a representation would generalize better compared than one learnt without the proposed method.

---

> ### Author Response · Authors · 2021-11-23
> **Author Response**
>
> Thank you so much for your review and the constructive comments! Please find our detailed response below:
>
> **I don't see why each predictor would not learn to filter out unnecessary information in the representation**: Consider an optimal (i.e. sufficient and minimal) representation to learn d features X. Now, when we train diverse predictors, each predictor learns certain features Xi, for i=1, …, 3, each corresponding to the set of spurious features for that particular predictor. However since there is only a single representation to support all predictor losses, this single representation should capture the features that are common across all predictors. Now if each of the superior feature sets are diverse enough, the common features are guaranteed to be that from X, i.e. the optimal feature set. This is essentially the overall reasoning here. Note that individual predictors will not throw away useless information since by definition this useless information or spurious features are useful for minimizing that predictor’s loss (hence the word spurious correlations). However, this same set may not be useful for the other predictors and therefore the only place they can be removed is where there is a bottleneck, i.e. the representation.
>
> **"Evaluating with a linear predictor"**: We have added the main citation for this (Tian et. al 2021), where the authors present a very good case for why the linear predictor setting is worth studying. We have also added results on the ImageNet dataset (shown in the common respsonse). As to why the performance lags for the non-linear predictor case for SimSiam, we repeat the reply to reviewer 8QAk below:
>
> *For the self-supervised vision experiments, the asymmetry of the base SimSiam architecture requires that the asymmetric predictor “keeps up” with the projector. Since in modinv we are using multiple projectors, the predictor is trained for all samples but only one projector is trained for a particular sample. This is why we see that the performance slightly lags with the base case. We do not see this for a symmetric architecture like Barlow Twins (BT), as is observed by our ImageNet results. In the future, testing ModInv alongside a symmetric architecture such as SIMCLR would make more sense, particularly because it does not use a sparsity inducing term like in Barlow twins which might diminish the gap between vanilla BT and BT + ModInv.*
>
> **Can you clarify : "Unlike the reinforcement learning case..."**: So because of the asymmetric nature of the projector-predictor layers in the SimSiam case (vision), it has been shown that the predictor needs to always keep up with the projectors for it to learn good representations. This is in contrast with the RL case, where we do not really need the reward decoder to keep up with the changing representation. This is what we meant by the above statement. Indeed for symmetric vision architectures, this is not the case and the predictor is only used to train the representation.
>
> Tian et. al 2021. Understanding self-supervised learning dynamicswithout contrastive pairs.

---

### Official Review · Reviewer_8QAk · 2021-11-04

**Correctness:** 3
**Technical Novelty And Significance:** 3
**Empirical Novelty And Significance:** 3
**Recommendation:** 5
**Confidence:** 4

**Main Review:**

I will now highlight the strengths and weaknesses of the paper. The strengths and weaknesses will highlight the quality, clarity, and significance of the proposed approach.

**Strengths**

1. The paper is generally well-written and easy to follow. With the exception of a few points covered under weaknesses, the authors do a good of introducing the proposed approach and walking the reader through different experimental settings.

2. On the DMC suite, ModInv seems to offer improvements over Dreamers, CURL, DBC in most cases (excluding RAD settings, please refer to the weaknesses). In vision settings, when using a linear predictor, ModInv seems to offer significant improvements on CIFAR-10 when compared to a recent self-supervised approach (SimSiam).

**Weaknesses**

1. I’m not sure I completely follow the thread of motivation introduced in the paper. ModInv is right now motivated from a sparsity/minimality point of view (in the context of how they’ve been described in the submission) and the reasoning for the particular instantiation relies on introducing invariance across multiple predictors which subsequently reduces redundancy in the learned representations. I’m not sure if this hop of reasoning works. It’s not entirely clear to me that being resilient to changes in downstream predictors will necessarily reduce redundancy in the learned representations. Can the authors comment on this? Additionally, I would suggest the authors also discuss [A], where the information bottleneck interpretation of self-supervised learning w.r.t. a downstream task has been studied, in related work and situate their proposed approach accordingly

2. The paper suffers from a few clarity issues. Highlighting them here. For instance, the ModInv structure implies that each predictor head is trained independently (a given sample only trains one head), implying that there is some underlying routing of the sample to a predictor head is going on. It’s unclear from the draft right now how this routing is performed. As in, given a training sample, how do the authors decide which head to feed the corresponding representation? If this routing is random, can the authors comment if they investigated other routing strategies? In Section 5.2, the authors mention that mean correlation across feature dimensions decreases more steadily when using ModInv. However, the draft doesn’t provide any supporting evidence (a table/plot) for this. More specifically, evidence demonstrating the evolution of mean correlation when using vanilla, ModInv, and an agent trained with mean correlation loss would help solidify this point even more. This is particularly important since the bulk of the motivation for ModInv relies on this being a desirable and observed behavior.

3. I think the experimental evidence in support of the proposed approach is somewhat weak. Firstly, ModInv + Aug outperforms RAD in only one setting (Reacher, Easy at 500k; Table 1). Did the authors use the same set of augmentations being used in RAD? If yes, then the utility of ModInv is heavily diminished. Particularly since ModInv involves using extra parameters. Results on computer vision benchmarks seem strong only for one setting — CIFAR-10 with a linear predictor. ModInv is competitive or worse on STL-10 and when a non-linear predictor is used, ModInv seems to consistently hurt performance in most settings. Furthermore, the experiments in Table 4 seem to suggest that ModInv outperforms Mean Correlation Loss in only one (out of two) settings. Given these observations, I am not entirely confident of the utility of the proposed approach. Additionally, I would suggest the authors appropriately adjust the claims regarding performance improvements. An ablation mentioned in Section 4.2.2 states that using individual SimSiam models with different learning rates is worse than using different predictor heads. The paper would benefit if this ablation was done in a more comprehensive manner, i.e., across both RL and vision settings at different stages. Since ModInv relies on using multiple predictor heads and using different models is a natural alternative, including such results seem necessary and would definitely improve the paper.

4. [Minor Points] Can the authors comment on how the learning rates and initializations for different predictor heads in all settings were chosen? Given that diversifying predictor heads is a big part of ModInv (as indicated from the ablations in Table 5), specifying these details seem important. In Section 6.1, the authors suggest that ModInv is similar to DIB in terms of instantiation. It’s unclear to me how that’s true. Can the authors comment on this?

[A] - Self-supervised Learning from a Multi-view Perspective

**Summary Of The Paper:**

Motivated from the idea that minimal representations — that encode information relevant to a task and nothing more — are likely to generalize well, the paper proposes an approach, titled Model Invariance (ModInv) to learn representations. The underlying idea is to use multiple diversified predictors solving auxiliary objectives on top of a shared representation/encoder. Different predictors are diversified in terms of their training dynamics — different initializations and learning rates. The approach relies on the hope that a shared representation that can lead to optimal performance for a diverse set of predictors is likely encoding some notion of implicit invariance to changes in predictors, thereby, introducing some notion of minimality. The authors conduct experiments on reinforcement learning and self-supervised (in computer vision) settings and claim that ModInv provides strong performance improvements in both. Based on further ablations,  the authors demonstrate how sensitive ModInv is to the number of predictors and the diversity in terms of training dynamics (in terms of learning rate), and whether ModInv is complementary to recent augmentation strategies in reinforcement learning.

**Summary Of The Review:**

The points highlighted under strengths and weaknesses form the basis of my rating. In particular, the weaknesses influence my current rating of the paper the most. While the paper is generally well-written, I think the issues highlighted under weaknesses in terms of motivation, clarity issues and lack of strong experimental evidence makes me less convinced of the utility of ModInv. I would encourage the authors to address the first three points under weaknesses. Addressing those will greatly help in reconsidering my rating of the paper. The minor points highlighted under 4 are addressable, I think.

---

> ### Author Response · Authors · 2021-11-23
> **Main Response**
>
> Thank you so much for your review and the constructive comments! Please find our detailed response below:
>
> **“introducing invariance across multiple predictors which subsequently reduces redundancy in the learned representations”**
>
> There are two ideas to unpack here. One is that multiple predictors over a common representation lead to a minimality objective. The other is that of reduced redundancy in the learned representations. You shouldn’t think that optimizing for the first would correspond to a lower redundancy per se. This is since lower redundancy (a.k.a our version of sparsity) corresponds to the entropy of the representation. Of course, zero entropy does not help learn an informative representation. However, a positive entropy does not particularly correspond to one of the desired terms exclusively, i.e. sufficiency or minimality. Nevertheless, it is indeed an interesting property to look at. Finally, for self-supervised vision methods and RL methods to some extent, an derivation of interest can be found in the Appendix section of the Barlow Twins paper.
>
> **Routing representations of data samples**: Yes, this is done by randomly choosing a predictor head all throughout the paper. We did not experiment with more sophisticated schemes but do think it is an interesting direction to improve along.
>
> **RL results**: We found a bug in the modinv + aug results for the RL experiments. As noted in the main comment, fixing these results in most scores remaining within 1 standard deviation of each other. Since the maximum score for DMC envs is 1000, there is very little ceiling for improvement of results with augmentations added for most tasks.
>
> **Performance for Vision methods**: For the self-supervised vision experiments, the asymmetry of the base SimSiam architecture requires that the asymmetric predictor “keeps up” with the projector. Since in modinv we are using multiple projectors, the predictor is trained for all samples but only one projector is trained for a particular sample. This is why we see that the performance slightly lags with the base case. We do not see this for a symmetric architecture like Barlow Twins (BT), as is observed by our ImageNet results. In the future, testing ModInv alongside a symmetric architecture such as SIMCLR would make more sense, particularly because it does not use a sparsity inducing term like in Barlow twins which might diminish the gap between vanilla BT and BT + ModInv.
>
> **Choosing the different learning rates**: For now, this is done simply by intuition, i.e. if the base lr is 3e-3 then we choose one lr slightly higher, i.e. 5e-3 and one lr slightly smaller, i.e. 1e-3. We believe in future work there is a lot of scope in developing better engineering solutions for such choices. We have noted this in the updated version.
>
> **DIB** uses the information bottleneck principle but for a particular family of predictors only. Here, we use the same idea (same architecture for all predictors) and thus it is closely related to DIB. Note that DIB was using fake labels and reversing gradients for the K predictors so as to enforce minimality. However, that creates a min-max optimization problem which could be hard to solve in practice. We don’t face that issue but of course need to pay in terms of more effort to make the predictors as diverse as possible.

---

> > ### Comment · Reviewer_8QAk · 2021-11-27
> > **Thanks for the response**
> >
> > Apologies for the delay and thanks to the authors for responding to my concerns. I read the other reviews and the corresponding responses to the same. Highlighting below which concerns (mine or otherwise) I think were sufficiently addressed.
> >
> > Thanks for providing the updated ImageNet results with the Barlow Twins base model and the supporting hypothesis. This helps demonstrate the case where ModInv actually helps when coupled with SSL approaches. I would encourage the authors to include both the BT and the SimSiam results in future versions of the draft (with exhaustive results) to provide a more complete picture.
> >
> > Thanks for identifying the bug with the ModInv + Aug experiments. While it’s better that ModInv + Aug is roughly (as stated by the authors) equivalent to RAD in terms of task performance, it still doesn’t make a very strong case for the use of ModInv since ModInv requires training with more parameters.
> >
> > Thanks for clarifying the low-level details I requested in my review. They definitely help in terms of providing more context. Additionally, thanks for providing the additional minimality claim experiment. I would encourage the authors to include the same in future versions of the draft.
> >
> > I think the reply by the authors is still missing a response to the point about lack of evidence regarding “mean correlation decreases more steadily using ModInv”. As stated in my review, specifically, evidence demonstrating the evolution of mean correlation when using vanilla, ModInv, and an agent trained with mean correlation loss seems particularly important since the bulk of the motivation for ModInv relies on this being a desirable and observed behavior. This seems to be missing at the moment.
> >
> > Regarding the response to the comment about “invariance and redundancy”, I am uncertain what conclusion I should draw based on the reply by the authors. Essentially, is the hop of reasoning not what I pointed out in the review? If not, could the authors clarify what the specific hop of reasoning here is. Additionally, while I thank the authors for providing a response to the point about the lack of a solid connection to the information bottleneck setup (as pointed out by other reviewers), I still don’t find the response entirely convincing.
> >
> > Overall, while I appreciate the response provided by the authors for the reviewer concerns, I still think the experimental support for the claims in the paper is somewhat weak. Additionally, as pointed out above, my concerns (as pointed out in the review have not been sufficiently addressed) — in particular, the point about the mean correlation experimental evaluation. Given this, I am inclined to stick to my original rating of the paper.

---

### Author Response · Authors · 2021-11-23
**New Results**

Thank you to all reviewers for their efforts in providing critical feedback and comments!

- **ImageNet results**: We test ModInv over the Barlow Twins base model, with K=3 heads. The learning rates of the 3 heads were chosen randomly, i.e. starting with the base lr, we chose a lr slightly higher and another slighlty lower. We did not optimize for the different learning rates or the number of heads. The following result is obtained for the first implementation of ModInv:

|       | ImageNet |
| ----------- | ----------- |
| Barlow Twins      |  72.8  |
| Barlow Twins + ModInv  |  **73.4**  |

- **Minimality Claim**: To show that ModInv learns minimal representations, we borrow the toy setup from Dubois et. al, where we train the model on the CIFAR+MNIST dataset which has CIFAR-10 images with MNIST digits overlayed on it. We first train using the CIFAR labels, freeze the representation, and then train a new predictor head over the frozen representation to predict the MNIST digits. A minimal representation would ignore the MNIST labels when being trained to predict the CIFAR classes. Therefore, a lower accuracy on the MNIST prediction task indicates learning a minimal representation. In the results below, we see that ModInv (K=3) indeed leads to lower accuracy, indicating that there is less spurious correlations (MNIST digits in this case) present in the learned representaion.

|       | CIFAR+MNIST (MNIST acc.) |
| ----------- | ----------- |
| Base Model      |  18.4  $\pm$ 0.2|
| Base Model + ModInv  |  **16.7**  $\pm$ 0.2|

Note that both models lead to the same accuracy on the CIFAR labels.

- **Minor Update to the RL results (with augs)**: We realized there was a small bug in the code which put the ModInv + Aug comparison with RAD on a slight disadvantage. We have fixed this. Note that with augmentations, most results are within one standard deviation of each other.

We hope these results and responses below help address the reviewers' main concerns and if so, they consider updating their review accordingly. Thank you!

---

### Decision · Program_Chairs · 2022-01-20

**Decision:**

Reject

**Comment:**

Although all reviewers had many positive comments on the paper, and the authors engaged nicely in the discussion period, at the moment there is a consensus among the reviewers that the central claims of the paper (related to minimal representations / information bottleneck) are not adequately supported by the current experiments. In particular, there were concerns that performance gains could be due to diversity of predictors, rather than minimal representations, which would need to be addressed. It's suggested that the reviewers take all of these comments and discussion into account when preparing a revised version of the paper.